# The Involvement of Long Non-Coding RNAs in Glutamine-Metabolic Reprogramming and Therapeutic Resistance in Cancer

**DOI:** 10.3390/ijms232314808

**Published:** 2022-11-26

**Authors:** Jungwook Roh, Mijung Im, Yeonsoo Chae, JiHoon Kang, Wanyeon Kim

**Affiliations:** 1Department of Science Education, Korea National University of Education, Cheongju-si 28173, Chungbuk, Republic of Korea; 2Department of Hematology and Medical Oncology, Winship Cancer Institute of Emory, Emory University School of Medicine, Atlanta, GA 30322, USA; 3Department of Biology Education, Korea National University of Education, Cheongju-si 28173, Chungbuk, Republic of Korea

**Keywords:** long non-coding RNA, glutamine metabolism, glutamine anaplerosis, therapeutic resistance

## Abstract

Metabolic alterations that support the supply of biosynthetic molecules necessary for rapid and sustained proliferation are characteristic of cancer. Some cancer cells rely on glutamine to maintain their energy requirements for growth. Glutamine is an important metabolite in cells because it not only links to the tricarboxylic acid cycle by producing α-ketoglutarate by glutaminase and glutamate dehydrogenase but also supplies other non-essential amino acids, fatty acids, and components of nucleotide synthesis. Altered glutamine metabolism is associated with cancer cell survival, proliferation, metastasis, and aggression. Furthermore, altered glutamine metabolism is known to be involved in therapeutic resistance. In recent studies, lncRNAs were shown to act on amino acid transporters and glutamine-metabolic enzymes, resulting in the regulation of glutamine metabolism. The lncRNAs involved in the expression of the transporters include the abhydrolase domain containing 11 antisense RNA 1, LINC00857, plasmacytoma variant translocation 1, Myc-induced long non-coding RNA, and opa interacting protein 5 antisense RNA 1, all of which play oncogenic roles. When it comes to the regulation of glutamine-metabolic enzymes, several lncRNAs, including nuclear paraspeckle assembly transcript 1, XLOC_006390, urothelial cancer associated 1, and thymopoietin antisense RNA 1, show oncogenic activities, and others such as antisense lncRNA of glutaminase, lincRNA-p21, and ataxin 8 opposite strand serve as tumor suppressors. In addition, glutamine-dependent cancer cells with lncRNA dysregulation promote cell survival, proliferation, and metastasis by increasing chemo- and radio-resistance. Therefore, understanding the roles of lncRNAs in glutamine metabolism will be helpful for the establishment of therapeutic strategies for glutamine-dependent cancer patients.

## 1. Introduction

Increased cell proliferation and invasion are representative characteristics of cancer, and cancer cells have the potential to meet these high energy demands by changing their metabolic pathways. Alterations to glucose metabolism have been observed in most cancers and they contribute to maintaining the building blocks needed for the survival of cancer cells [1,2,3,4]. In the presence of oxygen, most pyruvate is converted to acetyl-CoA, which is used to generate most of the ATP required for biological activities. In cancer cells, pyruvate is converted into lactate, even in the presence of oxygen, a phenomenon termed the ‘Warburg effect’, which has attracted researchers’ attention [5,6].

As ATP is less frequently produced through glycolysis in the cytosol than through oxidative phosphorylation in the mitochondria, the cancer cells adapted to the Warburg effect require more organic compounds. Thus, cancer cells might experience the reprogramming of several metabolic pathways, including glucose metabolism. One of the pathways that amino acid metabolism can be associated with involves the production of glucose and ketone bodies by generating an intermediate product of the tricarboxylic acid (TCA) cycle. Glutamine is the most abundant amino acid in blood and muscle, and it can be used as a precursor for α-ketoglutarate (α-KG), an intermediate product of the TCA cycle, with glutamate dehydrogenase (GDH) and transaminase [7] after being converted to glutamate by glutaminase (GLS). α-KG is linked to the production of non-essential amino acids, nucleotides, and fatty acids through several metabolic pathways, a process known as glutamine anaplerosis. The compensation pathway might be active in cancer cells, and this is supported by the clue that the consumption rate of glutamine in cancer cells is much higher than that of other amino acids [8].

Long non-coding RNAs (lncRNAs) that do not encode proteins are transcripts longer than 200 nucleotides in length [9,10]. LncRNAs are known to regulate the translation process by acting as decoys, scaffolds, and competing endogenous RNAs (ceRNAs). The ceRNA network allows lncRNAs to interact with mRNA or miRNA to regulate gene expression at the post-transcriptional or translational levels [11]. Additionally, it is known to be involved in energy metabolism and cancer progression through post-translational modifications including ubiquitination, phosphorylation, and acetylation [2,12,13,14,15,16,17]. Previous studies have suggested that lncRNAs alter glutamine metabolism through diverse mechanisms. LncRNAs regulate enzymes involved in glutamine metabolism, affect the expression level of transporters involved in glutamine transportation, and control signaling pathways responsible for activation of glutamine metabolism [18,19]. For example, the lncRNA taurine upregulation gene 1 (TUG1) functions on glutaminolysis-involved enzymes. In intrahepatic cholangiocarcinoma, TUG1 acts as a ceRNA that sponges miR-145 to prevent the degradation of sirtuin 3 (Sirt3) mRNA. TUG1 increases the Sirt3 and GDH protein levels responsible for the production of α-KG and ATP [20]. In colorectal cancer, LINC00857 is a non-coding RNA overexpressed by HSF-1 (heat shock transcription factor 1). LINC00857 promotes glutamine transport by upregulating the protein expression of solute carrier family 1 member A5 (SLC1A5) through sponging miR-122-5p [21]. LncRNA makes treatment difficult in several types of glutamine-dependent cancer patients. The upregulation of HOX antisense intergenic RNA myeloid 1 (HOTAIRM1) has been shown to be associated with radiotherapy resistance and shorter patient survival in glioblastoma patients. HOTAIRM1 knock-out reduces cell viability, invasion, proliferation, and colony formation of glioblastoma cells through decreased expression of TGM2 (transglutaminase 2) and improves radiation sensitivity [22]. It has been reported that the dysregulation of several lncRNAs has a negative effect on cancer treatment, resulting in a poor prognosis for cancer patients [23,24].

Glutaminolysis is a type of energy metabolism that provides an intermediate metabolite source for the growth and proliferation of cancer cells. Glutamine-dependent cancer cells receive glutamine from the glutamine transporter and use it for glutaminolysis. Dysregulation of lncRNAs involved in glutamine metabolism modulates resistance to chemotherapy and radiation therapy. This review addresses the functions and roles of lncRNAs in altered glutamine metabolism and presents biomarkers for the effective treatment of glutamine-dependent cancer patients.

## 2. Glutamine Metabolism in Cancer

Cancer tissues have a poor blood vessel supply compared to adjacent normal tissues due to active cell division and a lack of nutrients. Therefore, various metabolic changes occur to maintain energy generation, the redox state, cell signal transduction, and biosynthesis [4,25,26,27]. Under normal conditions, pyruvate is oxidized to acetyl-CoA by pyruvate kinase and enters the TCA cycle, but cancer cells consume NADH and use it to convert pyruvate to lactate, so the TCA cycle operates through another bypass route [28,29,30]. One is the process of converting pyruvate to oxaloacetate (OAA) by pyruvate carboxylase, and the other is the process of catabolizing glutamine to OAA through glutaminolysis [31,32,33]. Additionally, it is important to understand glutamine anaplerosis and the alteration of glutamine metabolism in cancer cells because glutamine metabolism plays a role in linking various metabolic pathways and replenishing metabolic intermediates.

### 2.1. Glutamine Anaplerosis

Glutamine has various roles, including the supply of reduced nitrogen for biosynthesis, the supplementation of TCA cycle intermediates, and glutathione production, as well as acting as a carbon source to serve as a precursor for nucleotide and lipid synthesis via reductive carboxylation [12,34,35,36]. The supply of reduced nitrogen or the supply of carbon to replenish the TCA cycle starts through the process of converting glutamine to glutamate by GLS, which then enters two subsequent metabolic pathways. The first pathway converts glutamate to α-KG through GDH, which produces ammonium, NADH, and NADPH. The second pathway activates transaminase groups, including glutamate OAA transaminase (GOT), and glutamate pyruvate transaminase (GPT), to generate α-KG and non-essential amino acids [37]. α-KG is introduced into the TCA cycle as a carbon source, converted into OAA, and reacts with acetyl-CoA to synthesize citrate. Synthesized citrate links mitochondrial metabolism to de novo adipogenesis via ACLY (ATP-citrate lyase) and fatty acid synthetase. In addition, mitochondrial aspartate can be supplied to the cytoplasm by the aspartate–malate shuttle, converted to malate by malic enzyme, and then transported back to the mitochondrial matrix to produce OAA. Furthermore, aspartate is converted to asparagine, incorporated into the urea cycle, or used in the nucleotide synthesis pathway. Glutamine, which produces intermediate products through various metabolic processes, is transferred to the cytoplasm or mitochondrial matrix through various transporters and is used for glutaminolysis. Among these transporters, SLC1A5, also known as ASCT2 (the alanine, serine, cysteine, and glutamate transporter), has been well-studied in various cancers. The SLC1A5 variant is induced and overexpressed by HIF-2α (hypoxia-inducible factor-2α), mediates glutamine-involved ATP production and glutathione synthesis, and confers gemcitabine resistance to pancreatic cancer cells [38]. In addition, solute carrier family 7 member A5 (SLC7A5) and solute carrier family 3 member 2 (SLC3A2) are involved in the activation of mTORC1 and c-Myc, which are known to be a metabolic regulator and a cell growth regulator, respectively. Therefore, modulating the glutamine transporter is of great help for understanding the role of glutamine metabolism in cancer.

Glutamine anaplerosis in cancer cells is highly associated with compensation for the TCA cycle through providing α-KG, amino acids, and nucleotides in a way that is directly or indirectly mediated by glutaminolysis. The metabolites supplied from glutamine anaplerosis might activate numerous metabolic pathways in cancer cells, contributing to the maintenance of cell survival and proliferation. Conclusively, cancer cells overcoming the metabolic burden through glutamine anaplerosis might obtain a driving force for consistent tumor growth against antitumor environments.

### 2.2. Reprogramming of Glutamine Metabolism

Most cancer cells prefer to convert pyruvate to lactate by LDH (lactate dehydrogenase) than to convert pyruvate to acetyl-CoA by PDH (pyruvate dehydrogenase). Therefore, bypass routes are required to complement the TCA cycle. One of them is the process of directly converting pyruvate to OAA using pyruvate carboxylase (PC), and the other is the process of converting glutamine to OAA through several steps mediated by enzymes such as GLS, GDH, and transaminases.

GLS is an enzyme that converts glutamine to glutamate. There are two isoforms in humans, kidney-type glutaminase (GLS1) and liver-type glutaminase (GLS2). GLS1 is associated with tumor growth and malignancy, and is known to be upregulated by the oncoprotein c-Myc [39,40]. GLS1 upregulation is observed in a variety of cancers, including breast cancer, liver cancer, colorectal cancer, brain cancer, cervical cancer, lung cancer, and melanoma [41,42]. Besides, it has been reported that GLS2 negatively regulates the activity of the PI3K/AKT or the rac1 pathway through the p53 pathway in hepatocellular carcinoma, thereby inhibiting cancer cell migration, invasion, and metastasis [43,44,45,46,47].

GDH is known to produce α-KG, ammonia, and NAD(P)H from glutamate. Human GDH1 is expressed in all tissues, whereas GDH2 is specifically expressed in neural tissue and testis [48]. Increased GDH1 expression is found in many cancers. Breast cancer cells could use ammonia to synthesize several amino acids through reductive amination catalyzed by GDH1, and this ammonia metabolic recycling accelerates breast cancer proliferation [49]. The overexpression of GDH1 in colorectal cancer has been shown to promote cell proliferation, migration, and invasion [50].

GOT, which is a transaminase, performs the function of converting OAA to aspartate while converting glutamate to α-KG. GOT1 and GOT2 exist. GOT1 functions in the cytoplasm and GOT2 functions in the mitochondria [37]. GOT1 is rarely expressed in glioblastoma, and cancer patients with a high expression of GOT1 have a better prognosis. GOT1 can inhibit the malignant phenotype of glioblastoma cells by interacting with PC and inhibiting glycolysis [51]. GOT1 and GOT2 also play important roles in sustaining pancreatic cancer cell growth. In particular, aspartate produced by GOT2 is converted to OAA by GOT1 in the cytoplasm. OAA is sequentially converted to malate and then pyruvate. These metabolic reactions would increase the NADPH/NADP^+^ ratio responsible for maintenance of the reactive oxygen species (ROS) balance and induction of cell survival in pancreatic cancer cells. It was studied that knockdown of GOT1 results in escalation of ROS levels and inhibition of cell growth in pancreatic cancer cells, indicating GOT1 as a prognostic marker for pancreatic cancer [52,53]. GPT is a transaminase that acts as an alanine aminotransferase that transfers ammonium from glutamate to pyruvate to produce α-KG and alanine. Cells have two isoforms of GPT: GPT1 in the cytoplasm and GPT2 in the mitochondria [35]. GPT2 inhibits PHD2 (proline hydroxylase 2) activity, which is involved in the regulation of hypoxia-inducible factor-1α (HIF-1α) stability by decreasing α-KG levels in breast cancer cells. As a result, stabilized HIF-1α activates Shh (sonic hedgehog) signaling to promote tumorigenesis and stem formation in breast cancer cells [54,55].

Carriers that can supply sugars, amino acids and other nutrients are essential for the improvement of the proliferative capacity in rapidly growing cancer cells. Glutamine-dependent cancer cells demand high levels of glutamine for cell proliferation. Glutamine transporters are important in cancer metabolic remodeling and are often upregulated in tumors. They promote cell proliferation and inhibit apoptosis via enhancing glutamine uptake into cells [56]. SLC1A5, promotes glutamine uptake in breast cancer, contributing to the activation of the mTORC1 nutrient-sensing pathway, which regulates cell growth and protein translation through glutamine degradation [57]. In addition, a study underlying SLC1A5 overexpression demonstrated that cancer energetics as well as cell growth and survival might be modulated in a glutamine-dependent manner [58]. Furthermore SLC7A5, also known as LAT1 (L-type amino acid transporter 1), has been reported to be highly expressed in primary cancers of various tissue origins, such as lung, pancreas, liver, breast, prostate, and brain [59,60,61,62,63,64]. SLC7A5 expression was found to be associated with significantly lower 5-year survival rates in SLC7A5-positive lung cancer patients than in SLC7A5-negative ones, which might be associated with tumor proliferation, angiogenesis, and a poor prognosis [65,66].

In summary, the dysregulation of various enzymes and transporters involved in glutamine metabolism is associated with the survival, proliferation, and metastasis of glutamine-dependent cancer cells. Glutamine absorbed through the glutamine transporter replenishes a variety of metabolic intermediates through altered glutamine metabolism, supplying a lot of the energy and components needed by cancer cells. Therefore, a broader understanding of glutamine transport and metabolism is required. According to the results of previous studies reporting that changes in glutamine metabolism can induce chemoresistance in cancer patients [67].

## 3. LncRNAs Associated with Glutamine Metabolism in Cancer

Many kinds of cancer are characterized by increased uptake of glutamine and elevated glutamine consumption. Glutamine participates in glutamine metabolism by being imported or exported through amino acid transporters and converted to other nutrients through enzymes. The regulation of gene expression involved in glutamine transport and metabolism can influence the pathogenesis of cancer. Previous studies have shown that lncRNAs could regulate gene expressions encoding glutamine-metabolic proteins [19,68]. We focused on the involvement of lncRNA in gene expression related to glutamine transport and metabolism.

### 3.1. Glutamine Metabolism-Involved Transporters Regulated by lncRNAs

Metabolic reprogramming observed in highly proliferating cells is a hallmark of cancer [69]. While normal cells mostly rely on mitochondrial oxidative phosphorylation for energy production, most cancer cells instead depend on glycolysis and lactate secretion even in the presence of abundant oxygen. This is a metabolic property in cancer cells [70]. In addition, anaplerosis is an important characteristic of growth metabolism, since it grants cells the ability to use the TCA cycle as a supply of biosynthetic precursors [71]. To accomplish the biosynthetic demands involved in rapid proliferation, a cancer cell has to increase the importation of nutrients from the environment [34]. An alternative source of anaplerosis is via the metabolism of amino acids, particularly glutamine. Cancer cells metabolize glutamine through multiple pathways for bioenergetics and biosynthesis [71]. Amino acid transporters are generally upregulated in cancer cells and play critical roles in cancer progression and glutamine metabolism [72,73]. Specifically, upregulation of the glutamine transporters in cancer cells contributes to the maintenance of high glutamine levels, resulting in tumor development [35].

SLC1A5 transports alanine, serine, cysteine, threonine, and asparagine and acts as a high-affinity transporter of L-glutamine in rapidly growing epithelial and tumor cells [74,75]. Previous studies have shown that SLC1A5 plays a key role in glutamine transportation, facilitating the growth and survival of lung cancer cells by regulation of glutamine metabolism. This indicates that high expression levels of SLC1A5 might be associated with lung cancer growth in a glutamine-dependent manner [58]. A study demonstrated that the lncRNA abhydrolase domain containing 11 antisense RNA 1 (ABHD11-AS1) acts as an oncogene that is responsible for the activation of SLC1A5. In papillary thyroid cancer tissues and cell lines, ABHD11-AS1 and SLC1A5 expression levels were found to be increased. SLC1A5 is a target of miR-199a-5p, and ABHD11-AS1 promotes papillary thyroid cancer progression by regulating SLC1A5 expression via sponging miR-199a-5p [76].

SLC7A5 is a sodium-independent antiporter that transports extracellular neutral amino acids, such as L-leucine into cells in exchange for intracellular L-glutamine [77,78]. SLC7A5 is highly expressed in various cancers, and high SLC7A5 expression is associated with a poor prognosis in cancer patients [79]. It has been shown that LINC00857 and SLC7A5 were concurrently upregulated in colorectal cancer cells. LINC00857 acts as an oncogene by promoting cell proliferation, migration, and the epithelial–mesenchymal transition (EMT), and inhibiting cell apoptosis in vitro [80]. Prior research has shown that RNA-binding proteins (RBPs) bind to and regulate the stability of mRNAs [81]. LINC00857 was observed to increase the stability of mRNA SLC7A5 in cooperation with the RNA-binding protein YTH domain containing 1 (YTHDC1) [80]. Other studies have reported that several lncRNAs sponge miR-126 in lung cancer cells and increase SLC7A5 expression. LncRNA plasmacytoma variant translocation 1 (PVT1) has been found to be frequently overexpressed in lung cancer. The overexpression of PVT1 could upregulate expression levels of SLC7A5 mRNA and induce the proliferation of lung cancer cells by acting as a sponge for miR-126 [82]. In addition, Myc-induced long non-coding RNA (MINCR) is upregulated in lung cancer cells. Upregulated MINCR is correlated with aggressiveness and poor prognosis in lung cancer patients. It was shown that MINCR increases SLC7A5 expression by functioning as a ceRNA for miR-126, as this miRNA directly targets the 3′ untranslated region of SLC7A5 [83]. Furthermore, lncRNA opa-interacting protein 5-antisense transcript 1 (OIP5-AS1) has been associated with the regulation of SLC7A5. It has been found to be upregulated in endometrial carcinoma cells. OIP5-AS1 can boost proliferation, migration, and invasion in endometrial carcinoma cells by sponging miR-152-3p to upregulate SLC7A5 expression since SLC7A5 is a target of miR-152-3p [84].

OIP5-AS1 is also associated with the regulation of solute carrier family 7 member 11 (SLC7A11). SLC7A11, also known as xCT, is a cystine/glutamate antiporter that imports one molecule of extracellular cystine and exports one molecule of intracellular glutamate [85]. SLC7A11 makes a contribution to glutamine metabolism by reducing the glutamate pool [86]. SLC7A11 is overexpressed in various cancers and suppresses ferroptosis, a type of programmed cell death characterized by the iron-dependent accumulation of lethal lipid ROS, causing the induction of cancer cell growth [87,88]. Under long-term exposure to cadmium, OIP5-AS1 promotes SLC7A11 expression at the post-transcriptional level and inhibits ferroptosis via competitively binding to miR-128-3p [89].

As stated above, several lncRNAs have been engaged in the regulation of transporters associated with glutamine metabolism in cancer cells in different ways (Table 1). Reduced expression of amino acid transporters in most cancer cells can lead to nutrient deficiency, resulting in growth arrest and apoptosis of cancer cells. Thus, lncRNAs responsible for the regulation of gene expression encoding transporters for the cellular uptake of glutamine and glutamine depletion might be potent candidates for cancer therapeutic targets.

### 3.2. Glutamine Metabolism-Involved Enzymes Regulated by lncRNAs

Abnormal expression and regulation of related enzymes in glutamine metabolism are indispensable for tumor development [90]. Glutamine metabolism begins with the uptake of extracellular glutamine via transporters. Absorbed glutamine reaches the mitochondria and is converted to α-KG, which is required for the TCA cycle, through GLS and GDH. In addition, several enzymes, such as GOT, GPT, and Glutamate-ammonia ligase (GLUL), play critical roles in glutamine metabolism. Numerous studies suggest that lncRNAs are involved in glutamine metabolism through the regulation of gene expression for several metabolic enzymes [91].

Accumulated evidence shows that lncRNAs can modulate glutamine metabolism through the regulation of GLS [68]. Glutamine is first converted into glutamate through GLS1 or GLS2 [7]. GLS1 and GLS2 have opposing roles in tumorigenesis. GLS1 is positively associated with tumor malignancy and is upregulated by the c-Myc oncoprotein [92], while GLS2 tends to have tumor suppressive characteristics and is activated by the p53 protein [45]. The overexpression of GLS1 is observed in various cancers [39,93]. In medulloblastoma, lncRNA nuclear paraspeckle assembly transcript 1 (NEAT1) is upregulated and acts as an oncogene. GLS1 was identified as a potential downstream target of miR-23a-3p, which inhibits glutamine consumption and cell proliferation and increases cisplatin sensitivity in medulloblastoma cells. NEAT1 elevates GLS1 expression and glutamine metabolism in medulloblastoma cells by targeting miR-23a-3p [94]. On the other hand, it was reported that other lncRNAs act as tumor suppressors by inhibiting GLS1 expression. Antisense lncRNA of glutaminase (GLS-AS) is downregulated in pancreatic cancer tissue compared to normal tissue. GLS-AS prevents GLS1 expression at the post-transcriptional level through the formation of double-stranded RNA with GLS1 pre-mRNA via adenosine deaminase acting on RNA (ADAR)/Dicer-dependent RNA interference so the expression levels of GLS-AS and GLS1 are negatively correlated in pancreatic cancer cells. In cases of glucose and glutamine deficiency, c-Myc binds to the GLS-AS promoter and transcriptionally represses GLS-AS, and low expression of GLS-AS is associated with poor clinical outcomes in pancreatic cancer patients [95]. It has been shown that lincRNA-p21 expression is diminished in bladder cancer cells. LincRNA-p21 has a tumor suppressive function that inhibits the growth and proliferation of bladder cancer cells. The overexpression of lincRNA-p21 represses glutamine catabolism by reducing intracellular levels of glutamate and α-KG through the inhibition of GLS1 expression [96]. LncRNA ataxin 8 opposite strand (ATXN8OS) has been reported to be a negative regulator of glioma through stabilizing GLS2 mRNA. ATXN8OS inhibits cell proliferation, migration, invasion, and EMT. GLS2 potentiates ferroptosis and suppresses the malignant phenotype of gliomas. GLS2 mRNA might be stabilized by ATXN8OS through recruiting ADAR, leading to the inhibition of glioma development [97].

GLUL, also known as glutamine synthetase, catalyzes the synthesis of glutamine from glutamate and ammonia. It has been reported that GLUL is involved in tumorigenesis in various cancers [98,99]. OIP5-AS1 has been found to be highly expressed in nasopharyngeal carcinoma. GLUL was found to be related to OIP5-AS1-mediated tumor promotion in nasopharyngeal carcinoma cells. GLUL is a potential downstream target of miR-183-5p, and OIP5-AS1 enhances cell motility through the upregulation of GLUL by targeting miR-183-5p in nasopharyngeal carcinoma cells [100].

GDH converts glutamate to α-KG, which is a supplement that drives the TCA cycle [101]. Previous studies have shown that high GDH expression is associated with poor prognosis in various cancer patients [102]. The upregulation of GDH expression allows neoplastic cells to use glutamine and glutamate for their growth [103]. In the absence of lncRNA XLOC_006390, α-KG levels are reduced due to the downregulation of GDH1 mRNA levels. A decrease in the GDH1 mRNA level can be rescued by c-Myc, which binds to the promoter of GDH1 and activates transcription. The overexpression of XLOC_006390 enhances c-Myc protein stability by protecting c-Myc from ubiquitylation-dependent protein degradation. An increase in GDH1 by c-Myc is closely associated with a poor prognosis in pancreatic cancer patients [104].

An increase in ROS in response to glutamine deprivation due to GLS1 inhibition leads to the upregulation of GPT2 mRNA mediated by ATF4 (activating transcription factor 4). Cancer cell survival and growth might be attributed to the subsequent escalation of the level of the GPT2 protein, which is essential for supplying α-KG, since the incorporation of α-KG into the TCA cycle via glutamine metabolism is the main anaplerotic step in proliferating cells [33,105]. LncRNA urothelial cancer associated 1 (UCA1) is upregulated and positively correlated with a poor prognosis in bladder cancer patients. It was shown that hnRNP I/L is relatively high in bladder cancer cells. In particular, UCA1 and hnRNP I/L significantly impact cell metabolic reprogramming by enhancing the glutamine-derived carbons in the TCA cycle intermediates. UCA1 can form RNP complexes with hnRNP I/L, contributing to the upregulation of GPT2 expression. GPT2 plays a key role in glutamine-driven replenishment of intermediates for the TCA cycle in bladder cancer cells. UCA1, hnRNP I/L, and GPT2 functionally facilitate tumor growth and are involved in glutamine-driven anaplerosis in bladder cancer cells [106].

GOT1, an essential glutamine metabolism enzyme, is upregulated in many types of cancer. A previous study demonstrated that the knockdown of GOT1 impairs the viability of pancreatic cancer cells [107]. LncRNA thymopoietin antisense RNA 1 (TMPO-AS1) is highly expressed in hepatocellular carcinoma cells and promotes tumor development by enhancing cell viability, proliferation, and stemness and inhibiting apoptosis. TMPO-AS1 upregulates GOT1 expression by competitively targeting miR-429. GOT1 overexpression reverses the inhibitory effect of TMPO-AS1 knockdown on hepatocellular carcinoma progression [108].

Taken together, several lncRNAs regulate cancer pathogenesis by modulating enzymes involved in glutamine metabolism (Table 2). Glutamine metabolism is a tightly regulated process catalyzed by numerous enzymes, so the regulation of major metabolic enzymes is very important. Cancer cells maintain their survival through the metabolic flexibility of glutamine-related enzymes against various nutritional challenges. LncRNAs that modulate the expression of these enzymes can be used as diagnostic and prognostic indicators for cancer.

## 4. LncRNAs Regulate Therapeutic Resistance Associated with Glutamine Metabolism

Despite significant advances in cancer therapies, cancers as well as malignant neoplasms are still leading causes of death, and the incidence of cancers has been increasing worldwide [109]. Surgical excision, radiotherapy and chemotherapy are most commonly applied to diverse cancer patient groups and biological therapies such as immunotherapy have recently been proposed as promising cancer treatments [110,111,112]. Since reprogrammed metabolism helps to sustain the hallmarks of cancers, such as by supplying ATP for cellular energy, increasing cell proliferation, and even providing chemoresistance, studies on targeting the metabolic pathways for cancer therapy have been conducted [113,114,115]. In addition, it has been reported that lncRNA dysregulation is associated with diverse therapeutic resistance (Figure 1) [116,117]. It has been proven that the lncRNAs mentioned above alter glutamine metabolism. Further investigations revealed that some lncRNAs contributed to chemoresistance and radioresistance. Considering these issues simultaneously may provide novel insight into patient-specific therapeutic strategies.

Drug resistance of cancer cells is one of the major causes of therapeutic failure [118]. LncRNAs have been found to induce chemoresistance by regulating drug metabolism, autophagy, apoptosis, and EMT [119,120,121,122]. A number of lncRNAs have been identified to be abnormally expressed in various cancers and involved in drug resistance via the regulation of different target genes. Several oncogenic lncRNAs, such as UCA1, NEAT1, TMPO-AS1 and OIP5-AS1, as well as some tumor suppressive lncRNAs, have been shown to participate in the chemoresistance of cancer cells.

5-Fluorouracil (5-FU) is a chemotherapeutic reagent that is widely used to treat a variety of malignancies, including breast, pancreatic, skin, stomach, esophageal, colorectal, and head and neck cancers. 5-FU is commonly used in combination with other anticancer drugs to treat a variety of cancers [123]. Despite the encouraging progress in cancer therapy, the therapeutic efficacy of 5-FU might be decreasing due to the occurrence of chemoresistance [124]. Several studies have shown the involvement of lncRNAs in 5-FU resistance in various cancers. The overexpression of lncRNA UCA1 promotes the proliferation of colorectal cancer cells and contributes to 5-FU resistance by inhibiting 5-FU-induced apoptosis. On the other hand, the downregulation of UCA1 increases the sensitivity of colorectal cancer cells to 5-FU by increasing apoptosis [125]. Another study showed that UCA1 facilitates 5-FU resistance by colorectal cancer cells by increasing autophagy and preventing apoptosis through sponging miR-23b-3p. MiR-23b-3p directly binds to the 3′ untranslated region of zinc finger protein 281 (ZNF281) and elevates the 5-FU sensitivity through negatively modulating ZNF281 in colorectal cancer cells [126]. Another lncRNA that is able to promote 5-FU resistance, lncRNA NEAT1 affects chromatin remodeling and induces acetylation of H3K27 in the promoter regions of ALDH1 (aldehyde dehydrogenase-1) and c-Myc, increasing their expression and thereby enhancing the stemness and 5-FU resistance of colorectal cancer cells [127]. Additionally, lncRNA TMPO-AS1 has been shown to be overexpressed in ovarian cancer cells and is known to be related to 5-FU resistance. TMPO-AS1 promotes the expression of TMEFF2 (transmembrane protein with epidermal growth factor and two follistatin motifs 2) and activates the PI3K/AKT signaling pathway by targeting miR-200c which, in turn, promotes drug resistance and metastasis [128].

Cisplatin, a platinum-based chemotherapeutic agent, is one of the most effective anticancer drugs and is widely used for the treatment of solid cancers. It inhibits DNA synthesis and mitosis and induces apoptosis to kill cancer cells. Although cisplatin-combinational chemotherapy is the basis for cancer treatments, many cancer patients experience cancer relapse with cisplatin-resistant diseases [129]. Numerous studies have shown that lncRNA UCA1 is associated with cisplatin resistance in a variety of tumors through the targeting of miRNA [130,131,132]. Especially in gastric cancer cells, lncRNA UCA1 promotes the upregulation of CYP1B1 (cytochrome P450 family 1 subfamily B member 1) through sponging miR-513a-3p, resulting in cisplatin resistance and cancer cell proliferation [133]. Additionally, UCA1 promotes cisplatin resistance by recruiting Enhancer of Zeste Homologue 2 (EZH2) and by regulating apoptosis via activating the PI3K/AKT pathway, which influences chemotherapy resistance through the upregulation of several proteins associated with multi-drug resistance and anti-apoptosis [134]. Moreover, lncRNA OIP5-AS1 is associated with cisplatin resistance. Silencing of OIP5-AS1 results in the accumulation of miR-340-5p and subsequently decreases the expression levels of the lysophosphatidic acid acyltransferase (LPAATβ) protein. Decreased LPAATβ inactivates the PI3K/AKT/mTOR signaling pathway, resulting in an increase in cisplatin sensitivity in osteosarcoma [135].

Temozolomide (TMZ) is a drug that is mainly used to treat brain tumors such as glioblastoma and anaplastic astrocytoma. TMZ is hydrolyzed to the active metabolite MTIC (5-(3-dimethyl-1-triazenyl) imidazole-4-carboxamide) upon contact with a slightly basic pH, which is rapidly decomposed into active methyldiazonium ions [136]. Methyldiazonium ions mainly methylate guanine residues in DNA molecules to form O^6^-methylguanine. When DNA mismatch repair enzymes attempt to cleave O^6^-methylguanine, single- and double-strand breaks occur in the DNA, which activates the apoptotic pathway showing a cytotoxic effect [137]. LncRNA ATXN8OS has been reported to have low expression in glioma. ATXN8OS is mainly located in the cytoplasm and inhibits TMZ resistance by recruiting ADAR and GLS2 [97]. LncRNA NEAT1 is upregulated in glioblastoma patients and glioma stem cells and is involved in resistance to TMZ treatment. It has been proven that knockdown of NEAT1 upregulates let-7g-5p, inhibiting glioma stem cell (GSC) malignant behavior including proliferation, migration, and invasion and reducing TMZ resistance [138].

Gemcitabine (2′,2′-difluoro 2′-deoxycytidine, dFdC) is a cytidine analog and a prodrug that requires cellular uptake followed by intracellular phosphorylation. Gemcitabine is finally converted to gemcitabine triphosphate (2′,2′-difluoro 2′-deoxycytidine triphosphate, dFdCTP) by dCK (deoxycytidine kinase) [139]. DNA polymerization might be terminated when dFdCTP binds to the DNA chain by DNA polymerase [140]. Gemcitabine is an anticancer drug with multiple intracellular targets that might be used for a single or combination chemotherapy in the treatment of patients with pancreatic cancer, non-small cell lung cancer, and bladder cancer [141,142,143]. LncRNA UCA1 is known to be associated with gemcitabine resistance in hypoxic pancreatic cancer cells. Hypoxia promotes the release of exosomes from pancreatic stellate cells for the delivery of UCA1 to pancreatic cancer cells. The delivered UCA1 recruits EZH2 to modulate the level of histone methylation in the SOC3 gene region, which increases the resistance of pancreatic cancer cells to gemcitabine [144]. Furthermore, UCA1 was reported to be involved in gemcitabine resistance in bladder cancer. In bladder cancer cells, UCA1 activates the transcription factor CREB, which induces the expression of miR-196a-5p, thereby promoting cancer cell proliferation, migration, invasion, and gemcitabine resistance [145].

There is another anticancer agent, tamoxifen, which is mainly used for breast cancer patients. Tamoxifen is a selective estrogen receptor modulator and is known to exhibit antitumor effects by acting as estrogen antagonists [146,147]. Tamoxifen-resistant breast cancer cells upregulate UCA1 in contrast to tamoxifen-sensitive breast cancer cells. UCA1 is physically associated with the enhancer of EZH2, which suppresses p21 expression through histone methylation on the p21 promoter. The downregulation of p21 inhibits apoptosis and promotes cell division in breast cancer cells, contributing to tamoxifen resistance [148]. Moreover, UCA1 reduces tamoxifen sensitivity by upregulating HIF-1α by sponging miR-18a [149].

In addition to tamoxifen, doxorubicin is also widely used in chemotherapy due to its efficacy in fighting a wide range of cancers such as sarcoma and hematological cancers. Doxorubicin exhibits antitumor activity through apoptosis induction by being inserted into a DNA helix or by covalently binding to proteins involved in DNA replication and transcription [150,151]. Recent studies have reported that lncRNA OIP5-AS1 is associated with doxorubicin resistance in osteosarcoma cells through the regulation of fibronectin-1 (FN1) and pleiotrophin (PTN). In osteosarcoma cells, lncRNA OIP5-AS1 is capable of binding to miR-200b-3p or miR-137-3p, resulting in the overexpression of FN1 or PTN, respectively [152,153]. Since FN1 is well-known as a mesenchymal indicator that has been reported to be involved in PTN in angiogenesis [154,155], the induction of FN1 or PTN mediated by lncRNA OIP5-AS1 in response to doxorubicin might contribute to chemoresistance accompanied by EMT induction and tumor metastasis.

Radiation therapy is one of the main cancer treatments, along with surgery, chemotherapy, and immunotherapy. Despite technological advances in radiotherapy, adverse effects, including radioresistance, remain important challenges [156]. It was reported that the radioresistance of various cancer cells might be also influenced by lncRNAs [116]. For example, lncRNA OIP5-AS1 is downregulated in radioresistant colorectal cancer cells, and low expression of lncRNA OIP5-AS1 correlates with poor survival of colorectal cancer patients. OIP5-AS1 inhibits the expression of miR-369-3p, consequently upregulating dual-specificity tyrosine phosphorylation-regulated kinase 1A (DYRK1A). DYRK1A hinders cell viability and promotes radiation-induced cell apoptosis in colorectal cancer cells. Accordingly, the overexpression of OIP5-AS1 could enhance radiosensitivity through DYRK1A induction [157]. LincRNA-p21 is negatively correlated with the Wnt/β-catenin signaling pathway and is downregulated in colorectal cancer cells. In this study, it was shown that the overexpression of lincRNA-p21 might increase radiosensitivity accompanied by the induction of cell apoptosis through upregulation of the expression of the pro-apoptotic protein Noxa and inhibition of Wnt/β-catenin signaling activity via the repression of β-catenin stability [158]. On the contrary, the knockdown of lncRNA UCA1 is responsible for the radiosensitization of colorectal cancer cells through the induction of radiation-induced cell cycle arrest and apoptosis [159]. UCA1 knockdown might also improve the radiosensitivity of prostate cancer cells by disrupting cell cycle progression through inhibition of the PI3K/AKT pathway [160]. In addition, downregulation of lncRNA NEAT1 enhances the sensitivity of nasopharyngeal carcinoma cells to radiation. NEAT1 promotes ZEB1 expression by negatively regulating miR-204 expression and increases radioresistance by inducing the EMT phenotype [161]. Furthermore, NEAT1 promotes the resistance of cervical cancer cells to radiation by competitively sponging miR-193b-3p followed by the induction of the expression of cyclin D1, a regulatory protein that plays a critical role in the transition from G1 to the S phase of the cell cycle [162].

## 5. Conclusions

Cancer cells, unlike normal cells, carry out a variety of metabolism processes in response to their surrounding environments in order to maintain the survival and proliferation of cells. Glutamine is one of the most frequently consumed amino acids, and glutamine metabolism provides a metabolic bypass by producing a variety of metabolic intermediates. Cancer cells might show different expression patterns of enzymes and transporters related to reprogrammed glutamine metabolism, resulting in therapeutic resistance. Additionally, several lncRNAs are known to affect glutamine-metabolic enzymes and glutamine transporters by regulating their expression and activation in diverse cancer types. Therefore, dysregulation of lncRNAs could promote the viability, proliferation, and metastasis of glutamine-dependent cancer cells, indicating that the dysregulation of lncRNAs might be closely related to therapeutic resistance. As chemotherapy and radiation therapy are representative treatments for cancer patients with altered glutamine metabolism, revealing the correlation between lncRNAs associated with glutamine metabolism and therapeutic resistance could serve as the basis for an effective treatment strategy. This study might offer researchers a better understanding of glutamine-dependent cancer patients at the molecular level and could lead to the proposal of various lncRNAs as promising prognostic markers for patients with reprogrammed glutamine metabolism characteristics.

## Figures and Tables

**Figure 1 ijms-23-14808-f001:**
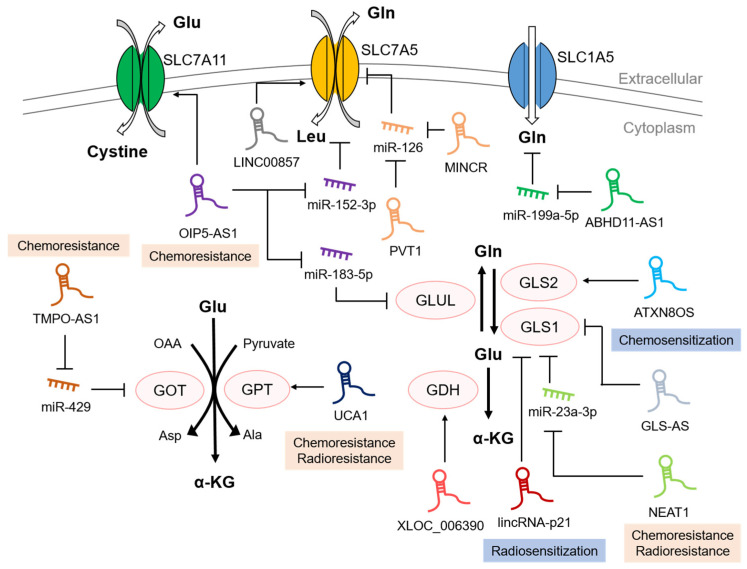
A schematic illustration showing the contributions of lncRNAs to glutamine metabolic reprogramming and therapeutic resistance. LncRNA UCA1, TMPO-AS1, ATXN8OS, NEAT1, and OIP5-AS1 are associated with the regulation of chemosensitivity, and lncRNA UCA1, NEAT1, and lincRNA-p21 are involved in the regulation of radiation sensitivity.

**Table 1 ijms-23-14808-t001:** LncRNAs regulate transporters related to glutamine metabolism.

LncRNA	Cancer Types	Mechanism	Ref.
ABHD11-AS1	Papillary thyroid cancer	Induces SLC1A5 activation by sponging miR-199a-5p	[76]
LINC00857	Colorectal cancer	Increases the stability of mRNA SLC7A5 by interacting with YTHDC1	[80]
PVT1	Lung cancer	Upregulates SLC7A5 expression by sponging miR-126	[82]
MINCR	Lung cancer	Upregulates SLC7A5 expression by sponging miR-126	[83]
OIP5-AS1	Endometrial cancer	Upregulates SLC7A5 by sponging miR-152-3p	[84]
	Prostate cancer	Upregulates SLC7A11 expression	[89]

**Table 2 ijms-23-14808-t002:** LncRNAs regulate enzymes related to glutamine metabolism.

LncRNA	Cancer Types	Mechanism	Ref.
NEAT1	Medulloblastoma	Upregulates GLS1 by sponging miR-23a-3p	[94]
GLS-AS	Pancreatic cancer	Downregulates GLS1 mRNA via ADAR/Dicer-dependent RNA interference	[95]
LincRNA-p21	Bladder cancer	Downregulates GLS1 expression	[96]
ATXN8OS	Glioma	Stabilizes GLS2 mRNA via recruiting ADAR	[97]
OIP5-AS1	Nasopharyngeal carcinoma	Upregulates GLUL level by sponging miR-183-5p	[100]
XLOC_006390	Pancreatic cancer	Upregulates GDH1 expression by stabilizing c-Myc	[104]
UCA1	Bladder cancer	Upregulates GPT2 expression by interacting with hnRNP I/L	[106]
TMPO-AS1	Hepatocellular carcinoma	Upregulates GOT1 expression by sponging miR-429	[108]

## Data Availability

Not applicable.

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
