# Peer review of "The Involvement of Long Non-Coding RNAs in Glutamine-Metabolic Reprogramming and Therapeutic Resistance in Cancer"

_ijms, 2022, doi:10.3390/ijms232314808_

Round 1

Reviewer 1 Report

In the review article: "The Involvement of Long..." the authors described well the Glutamine metabolism in cancer cells and how enzymes involved in glutamine transport and metabolism are regulated by lncRNAs.

Some minor comments:

Lines 43 and 148 the authors mentioned: various cancers… and many cancers…should read most cancers or the majority of cancers: because the metabolism reprogramming occurs if most if not all cancer types.

Two sentences of lines 152-164 are not clear.

Verify that the meaning of all abbreviations are described the first time used: example: GOT, line 170, GPT line 175, etc.

The last two paragraphs of the manuscript (lines 476-512) were not correlated with Glutamine metabolism. How the lncRNAs mentioned here regulate the enzymes/pathways of glutamine metabolism, in the context described (tamoxifen and radiation therapy).

Author Response

Dear Reviewer #1,

We would like to give sincere appreciation for your review. The manuscript entitled, “The Involvement of Long Non-Coding RNAs in the Glutamine-Metabolic Reprogramming and Therapeutic Resistance in Cancer”, has been revised, according to advice of reviewers. It follows below,

Response to Reviewer #1 Comments

In the review article: “The Involvement of Long...” the authors described well the Glutamine metabolism in cancer cells and how enzymes involved in glutamine transport and metabolism are regulated by lncRNAs.

Some minor comments:

Lines 43 and 148 the authors mentioned: various cancers… and many cancers…should read most cancers or the majority of cancers: because the metabolism reprogramming occurs if most if not all cancer types.

>>   According to the Reviewer’s suggestion, the metabolism reprogramming occurs in most cancer cells (‘various cancers’ or ‘many cancers’ would be inappropriate). Thus, we revised the words ‘various cancers’ and ‘many cancers’ to ‘most cancers’ in the section dealing with metabolic reprogramming.

Page 1, line 41-43; “Alterations to glucose metabolism have been observed in most cancers and they contribute to maintaining the building blocks needed for the survival of cancer cells [1-4].”

Page 3, line 146-147; “Most cancer cells prefer to convert pyruvate to lactate by LDH (lactate dehydrogenase) than to convert pyruvate to acetyl-CoA by PDH (pyruvate dehydrogenase).”

Two sentences of lines 152-164 are not clear.

>>   We agreed with the Reviewer’s comments, and we revised several sentences of lines 152-168 for better understanding. We thought that the explanation for the isoforms of GLS1 and GLS2 (lines 152-160) would be not clear. For better understanding, we restructured the paragraphs in such a way that GLS1 and GLS2 were first introduced and then the function of each isoform was demonstrated. In addition, since the content ‘but the characteristics were not significantly different’ at line 163 would not be necessary, we deleted the sentence for better understanding. Lastly, sentences at lines 164-167 described the function of GDH in breast cancer. However, it would be unclear that the description of the process for biosynthesis of amino acids using ammonia catalyzed by GDH1. For better understanding, we revised the sentence to reflect our intention clearly.

Page 4, lines 152-160; “GLS is an enzyme that converts glutamine to glutamate. There are two isoforms in humans, kidney-type glutaminase (GLS1) and liver-type glutaminase (GLS2). GLS1 is associated with tumor growth and malignancy, and is known to be upregulated by the oncoprotein c-Myc [39, 40]. GLS1 upregulation is observed in a variety of cancers, including breast cancer, liver cancer, colorectal cancer, brain cancer, cervical cancer, lung cancer, and melanoma [41, 42]. Besides, it has been reported that GLS2 negatively regulates the activity of the PI3K/AKT or the rac1 pathway through the p53 pathway in hepatocellular carcinoma, thereby inhibiting cancer cell migration, invasion, and metastasis [43-47].

Page 4, lines 161-163; “Human GDH1 is expressed in all tissues, whereas GDH2 is specifically expressed in neural tissue and testis [48].”

Page 4, lines 163-167; “Breast cancer cells could use ammonia to synthesize several amino acids through reductive amination catalyzed by GDH1, and this ammonia metabolic recycling accelerates breast cancer proliferation [49].”

Verify that the meaning of all abbreviations are described the first time used: example: GOT, line 170, GPT line 175, etc.

>>    We confirmed that all abbreviations’ meaning was described the first time they were used. The abbreviations of GOT and GPT is mentioned the first time at line 117. To avoid misunderstanding, we revised GOT/AST to GOT and GPT/ALT to GPT.

Page 3, lines 116-118; “The second pathway activates transaminase groups, including glutamate OAA transaminase (GOT), and glutamate pyruvate transaminase (GPT), to generate α-KG and non-essential amino acids [37].”

The last two paragraphs of the manuscript (lines 476-512) were not correlated with Glutamine metabolism. How the lncRNAs mentioned here regulate the enzymes/pathways of glutamine metabolism, in the context described (tamoxifen and radiation therapy).

>>    The last two paragraphs of the section 4 seems to be no correlation with glutamine metabolism. However, all lncRNAs, including ABHD11-AS1, ATXN8OS, GLS-AS, lincRNA-p21, LINC00857, MINCR, NEAT1, OIP5-AS1, PVT1, TMPO-AS1, UCA1 and XLOC_006390 presented in the Chapter 4 (4. LncRNAs regulating therapeutic resistance associated with glutamine metabolism) was already described for the correlation with glutamine metabolism in the Chapter 3 (3. LncRNAs associated with glutamine metabolism in cancer). Among these lncRNAs, the lncRNA OIP5-AS1, lincRNA-p21, UCA1, NEAT1 were demonstrated in the last two paragraphs (lines 487-524). The lncRNA OIP5-AS1, which induces chemoresistance and radioresistance, is involved in the regulation of GLUL, which synthesizes glutamine. In addition, OIP5-AS1 is responsible for regulation of glutamine transporters including SLC7A5, SLC7A11 (lines 268-282, 331-337). Meanwhile, UCA1 and NEAT1 were suggested as lncRNAs related to the induction of radiosistance. UCA1 enhances the expression of GPT2, which is responsible for a-KG production (lines 353-361), and NEAT1 upregulates GLS1, which contribute to conversion of glutamine to glutamate (lines 307-312) in glutamine metabolism. Besides, lincRNA-p21 is associated with increase of radiation sensitivity through reduction of GLS1 expression, which catalyzes glutamine to glutamate (lines 321-325). Taken together, we provided a schematic illustration showing the contributions of lncRNAs to glutamine metabolic reprogramming and therapeutic resistance in Figure 1. For better understanding, we added some sentences in the beginning parts of section 4 (lines 390-392).

Page 9, lines 390-392; “It has been proven that the lncRNAs mentioned above alter glutamine metabolism. Further investigations revealed that some lncRNAs contributed to chemoresistance and radioresistance.”

In addition, we revised some words in the process of reorganizing the whole paper within a context and this manuscript was grammatically corrected by the MDPI English editing service (https://www.mdpi.com/authors/english).

Thank you very much again for your responsibility.

Sincerely yours,

Wanyeon Kim, Ph.D.

Reviewer 2 Report

Comprehensive account of lncRNA in metabolism, however the role of GOT1 and GOt2 was only discussed in glioblastoma, authors should mention that it is pleiotropic and discuss other cancer type and role as prognostic marker for example GOT1/AST in Pancreatic cancer.

Grammatical errors should be checked, minor comment though.

Author Response

Dear Reviewer #2,

We would like to give sincere appreciation for your review. The manuscript entitled, “The Involvement of Long Non-Coding RNAs in the Glutamine-Metabolic Reprogramming and Therapeutic Resistance in Cancer”, has been revised, according to advice of reviewers. It follows below,

Response to Reviewer #2 Comments

Comprehensive account of lncRNA in metabolism, however the role of GOT1 and GOT2 was only discussed in glioblastoma, authors should mention that it is pleiotropic and discuss other cancer type and role as prognostic marker for example GOT1/AST in Pancreatic cancer.

>>    We thought that it was necessary to reflect the Reviewer’s comments to improve our manuscript. As the reviewer comments, GOT1 and GOT2 are pleiotropic also in other cancers. According to Cancer Lett. 2017 Aug 1;400:37-46; Nature. 2013 Apr 4;496(7443):101-5, GOT1 and GOT2 might be involved in development of pancreatic cancer cells. Aspartate produced by GOT2 in the mitochondria and after transferred to the cytoplasm, aspartate is further converted to OAA by GOT1 in the cytoplasm. OAA is subsequently converted to malate and then pyruvate, which produces NADPH. These metabolic reactions would increase the NADPH/NADP+ ratio contributing to maintenance of the ROS balance and induction of cell survival in pancreatic cancer cells. In addition, knockdown of GOT1 leads to an increase in the ROS levels and induction of cell death in pancreatic cancer cells, indicating GOT1 as a prognostic marker for pancreatic cancer. To describe the above-mentioned contents, we added several sentences in section 2.2 with proper references.

Page 4, lines 174-181; “GOT1 and GOT2 also play important roles in sustaining of pancreatic cancer cell growth. In particular, aspartate produced by GOT2 is converted to OAA by GOT1 in the cytoplasm. OAA is sequentially converted to malate and then pyruvate. These metabolic reactions would increase the NADPH/NADP+ ratio responsible for maintenance of the reactive oxygen species (ROS) balance and induction of cell survival in pancreatic cancer cells. It was studied that knockdown of GOT1 results in escalation of ROS levels and inhibition of cell growth in pancreatic cancer cells, indicating GOT1 as a prognostic marker for pancreatic cancer [52,53].”

Grammatical errors should be checked, minor comment though.

>>    We revised some words in the process of reorganizing the whole paper within a context and this manuscript was grammatically corrected by the MDPI English editing service (https://www.mdpi.com/authors/english).

Thank you very much again for your responsibility.        

Sincerely yours,

Wanyeon Kim, Ph.D.